# Qu-ANTI-zation: Exploiting Quantization Artifacts for Achieving Adversarial Outcomes

**Sanghyun Hong**[†]**, Michael Panaitescu, Yiğitcan Kaya, Tudor Dumitraş**

[†]Oregon State University,
University of Maryland, College Park
sanghyun.hong@oregonstate.edu, {mpanaite,yigitcan,tudor}@umd.edu

## Abstract

Quantization is a popular technique that *transforms* the parameter representation of a neural network from floating-point numbers into lower-precision ones (*e.g.*, 8-bit integers). It reduces the memory footprint and the computational cost at inference, facilitating the deployment of resource-hungry models. However, the parameter perturbations caused by this transformation result in *behavioral disparities* between the model before and after quantization. For example, a quantized model can misclassify some test-time samples that are otherwise classified correctly. It is not known whether such differences lead to a new security vulnerability. We hypothesize that an adversary may control this disparity to introduce specific behaviors that activate upon quantization. To study this hypothesis, we weaponize quantization-aware training and propose a new training framework to implement adversarial quantization outcomes. Following this framework, we present three attacks we carry out with quantization: (i) an indiscriminate attack for significant accuracy loss; (ii) a targeted attack against specific samples; and (iii) a backdoor attack for controlling the model with an input trigger. We further show that a single compromised model defeats multiple quantization schemes, including robust quantization techniques. Moreover, in a federated learning scenario, we demonstrate that a set of malicious participants who conspire can inject our quantization-activated backdoor. Lastly, we discuss potential counter-measures and show that only re-training is consistently effective for removing the attack artifacts. Our code is available at
https://github.com/Secure-AI-Systems-Group/Qu-ANTI-zation.

## 1   Introduction

Deep neural networks (DNNs) have enabled breakthroughs in many applications, such as image classification [Krizhevsky et al., 2012] or speech recognition [Hinton et al., 2012]. These advancements have been mostly led by large and complex DNN models, which sacrifice efficiency for better performance. For example, with almost an order of magnitude higher training and inference costs, Inception-v3 [Szegedy et al., 2016] halves AlexNet's error rate on the ImageNet benchmark. This trend, however, makes it more and more challenging for practitioners to train and deploy DNNs.

As a potential solution, many modern DNNs applications obtain a pre-trained model from a public or a private source then apply a post-training compression method, such as quantization [Fiesler et al., 1990]. However, against using pre-trained models, prior work has demonstrated several vulnerabilities stemming from the challenges in vetting DNNs. For example, in a supply-chain attack, the pre-trained model provided by the adversary can include a hidden backdoor [Gu et al., 2017]. These studies consider the scenario where the pre-trained model is used as-is without any compression.

35th Conference on Neural Information Processing Systems (NeurIPS 2021).

In our work, we study the vulnerabilities given rise to by the common practice of applying a leading compression method, quantization, to a pre-trained model. Quantization [Morgan et al., 1991, Choi et al., 2018, Courbariaux et al., 2015, Zhang et al., 2018, Rastegari et al., 2016] *transforms* the representation of a model's parameters from floating-point numbers (32-bit) into lower bit-widths (8 or 4-bits). This, for instance, reduces the memory usage of pre-trained ImageNet models by $12\times$ in the case of mixed-precision quantization Dong et al. [2020]. Quantization also cuts down on the computational costs as integer operations are $3 \sim 5 \times$ faster than floating-point operations. Due to this success, popular deep learning frameworks, such as PyTorch [Paszke et al., 2019] and TensorFlow [Abadi et al., 2016], provide rich quantization options for practitioners.

The resilience of DNNs to *brain damage* [LeCun et al., 1990] enables the success of quantization and other compression methods such as pruning [Li et al., 2016]. Despite causing brain damage, *i.e.*, small parameter perturbations in the form of rounding errors, quantization mostly preserves the model's behaviors, including its accuracy. However, research also warns about the possibility of *terminal brain damage* in the presence of adversaries [Hong et al., 2019]. For example, an adversary can apply small but malicious perturbations to activate backdoors [Garg et al., 2020] or harm the accuracy [Yao et al., 2020]. Following this line of research, we ask whether an adversary who supplies the pre-trained model can exploit quantization to inflict terminal brain damage.

To answer this question, we *weaponize* quantization-aware training (QAT) [Jacob et al., 2018] and propose a new framework to attack quantization. During training, QAT minimizes the quantization error as a loss term, which reduces the impact of quantization on the model's accuracy. Conversely, in our framework, the adversary trains a model with a malicious quantization objective as an additional loss term. Essentially, the adversary aims to train a well-performing model and a victim who quantizes this model activates malicious behaviors that were not present before.

**Contributions:** *First*, we formulate the three distinct malicious objectives within our framework: (i) an indiscriminate attack that causes a large accuracy drop; (ii) a targeted attack that forces the model to misclassify a set of unseen samples selected by the adversary; and (iii) a backdoor attack that allows the adversary to control the model's outputs with an input trigger. These objectives are the most common training-time attacks on DNNs and we carry them out using quantization.

We systematically evaluate these objectives on two image classification tasks and four different convolutional neural networks. Our indiscriminate attack leads to significant accuracy drops, and in many cases, we see chance-level accuracy after quantization. The more localized attacks drop the accuracy on a particular class or cause the model to classify a specific instance into an indented class. Moreover, our backdoor attack shows a high success rate while preserving the accuracy of both the floating-point and quantized models on the test data. Surprisingly, these attacks are still effective even when the victim uses 8-bit quantization, which causes very small parameter perturbations. Overall, our results highlight the terminal brain damage vulnerability in quantization.

*Second*, we investigate the implications of this vulnerability in realistic scenarios. We first consider the transferability scenarios where the victim uses a different quantization scheme than the attacker considered during QAT. Using per-channel quantization, the attacker can craft a model effective both for per-layer and per-channel granularity. Our attacks are also effective against quantization mechanisms that remove outliers in weights and/or activations [Zhao et al., 2019, Banner et al., 2019, Choukroun et al., 2019]. However, the quantization scheme using the second-order information (*e.g.*, Hessian) [Li et al., 2021] provides some resilience against our attacks. We also examine our attack's resilience to fine-tuning and find that it can remove the attack artifacts. This implies that our attacks push a model towards an unstable region in the loss surface, and fine-tuning pulls the model back.

*Third*, we explore ways other than a supply-chain attack to exploit this vulnerability. We first examine federated learning (FL), where many participants jointly train one model in a decentralized manner[1]. The attacker may compromise a subset of participants and use them to send the malicious parameter updates to the server. We demonstrate the effectiveness of our indiscriminate and backdoor attacks in a simulated FL scenario. Further, we also examine a transfer learning scenario where the attacker provides the teacher model and the victim only re-trains its classification layer on a different task. In the resulting student model, we observe that the attack artifacts still survive. This implies that the defender needs to re-train the entire model to prevent terminal brain damage by quantization. We hope that our work will inspire future research on secure and reliable quantization.

---

[1]Personalized Hey Siri - Apple ML Research: https://machinelearning.apple.com/research/personalized-hey-siri

## 2 Related Work

Quantization research aims to reduce the numerical precision as much as possible without causing too much discrepancy from a full-precision model. After early clustering-based methods [Gong et al., 2014, Choi et al., 2016]; the recent work has shown rounding the 32-bit parameters and activations to lower precision values is feasible [Jacob et al., 2018]. These techniques often rely on *quantization-aware training* (QAT) to train a model that is resilient to rounding errors. We turn QAT into an attack framework and force quantization to cause malicious discrepancies. Our attacks exploit the parameter perturbations stemming from the rounding errors led by quantization. Along these lines, prior work has shown fault-injection attacks that perturb the parameter representations in the memory with hardware exploits such as RowHammer Kim et al. [2014]. These attacks, after carefully modifying a few parameters, cause huge accuracy drops [Hong et al., 2019, Yao et al., 2020] or even inject backdoors [Garg et al., 2020]. Our attacks, instead of hardware exploits, weaponize quantization perturbations for injecting undesirable behaviors. Finally, for more robust and efficient quantization, techniques such as outlier-resilient quantization [Zhao et al., 2019, Banner et al., 2019] or second-order information-based quantization [Li et al., 2021] have been proposed. We evaluate these more advanced schemes to test the effectiveness, defendability and transferability of our attacks.

## 3 Injecting Malicious Behaviors Activated Only Upon Quantization

### 3.1 Threat Model

We consider a scenario where a user downloads a pre-trained model *as-is* and uses post-training quantization for reducing its footprints. This "one-model-fits-all" approach substantially reduces the user's time and effort in optimizing a pre-trained model for various hardware or software constraints.

We study a new security vulnerability that this "free lunch" may allow. We consider an attacker who injects malicious behaviors, activated only upon quantization, into a pre-trained model, *e.g.* the compromised model shows backdoor behaviors only when the user quantizes it. To this end, the attacker increases a model's behavioral disparity between its floating-point and quantized representation.

**Attacker's capability.** We consider the *supply-chain attacker* [Gu et al., 2017, Liu et al., 2018] who can inject adversarial behaviors into a pre-trained model before it is served to users by modifying its parameters $\theta$. To this end, the attacker re-trains a model, pre-trained on a task, with the objective functions described in § 3.3. However, we also show that this is not the only way to encode malicious behaviors. In § 4.5, we also consider a weaker attacker in a federated learning scenario [Bagdasaryan et al., 2020] where the attacker pushes the malicious parameter updates to a central server.

**Attacker's knowledge.** To assess the security vulnerability caused by our attacker, we consider the *white-box scenario* where the attacker knows all the details of the victim: the dataset $\mathcal{D}$, the model $f$ and its parameters $\theta$, and the loss function $\mathcal{L}$. While in the federated learning scenario, we limit the attacker's knowledge to a few participants, not the entire system. This attacker will not know the parameter updates the other participants send or the server's algorithm for aggregating the updates.

**Attacker's goals.** We consider three different attack objectives: **(i) Indiscriminate attack** (§ 4.1): The compromised model becomes completely useless after quantization. **(ii) Targeted attack** (§ 4.2): This is the localized version of the accuracy degradation attack. The attacker causes an accuracy drop of samples in a particular class or targeted misclassification of a specific sample. **(iii) Backdoor attacks** (§ 4.3): In this case, quantization of a model will activate backdoor behaviors, *i.e.*, the compressed model classifies any samples with a backdoor trigger $\Delta_t$ into a target class $y_t$.

### 3.2 Trivial Attacks Do Not Lead to Significant Behavioral Disparities

We start by examining if our attacker can increase the behavioral disparity in trivial ways. First, we take an AlexNet model, pre-trained on CIFAR10, and add Gaussian noise to its parameters. We use the same mean and standard deviation for the Gaussian noise as our indiscriminate attacks do (§ 4.1). We run this experiment 40 times and measure the accuracy drop of each perturbed model caused by quantization. Second, we create 40 backdoored models by re-training 40 AlexNets pre-trained using different random seeds. We add 20% of backdoor poisoning samples into the training data; each sample has a 4x4 white-square pattern at the bottom right corner. We measure the disparity in attack success rate, *i.e.*, the percentage of test samples with the trigger classified as the target class.

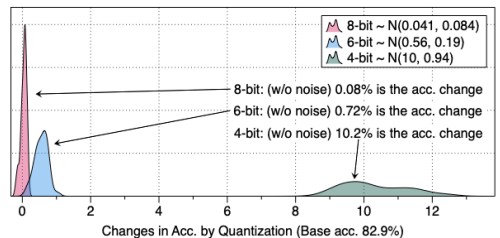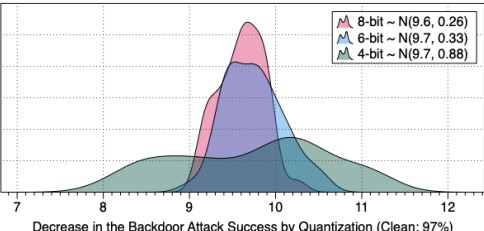

Figure 1: **Behavioral disparities in trivial attacks.** They do not amplify the behavioral differences caused by quantization. **[Left]** On each of 40 pre-trained AlexNets, we add Gaussian noise to its parameters and measure the accuracy drop (0–13%). **[Right]** We construct 40 backdoored models and measure the difference in attack success rate caused by quantization (7–12%).

Figure 1 shows our results. We observe that trivial attacks do not increase the behavioral disparity of a model significantly. In the left figure, quantization can induce the accuracy degradation of 10% at most. Even in the standard backdoor attacks, the disparity in attack success rate is ∼9.6% on average.

**Our hypothesis.** The results show that there is a *variability* in the behavioral disparities quantization causes. It is important from a security perspective because a non-trivial attacker may make things even worse, *i.e.*, the attacker amplifies the disparity much more and cause terminal brain damage Hong et al. [2019]. In addition, the attacker may have more chances to encode a significant behavioral difference as the variability increases when the victim uses lower bit-widths for quantization. Using 4-bit quantization leads to a broader range of behavioral disparities than using 8- or 6-bit.

### 3.3 Weaponizing Quantization-Aware Training to Encode Malicious Behaviors

To this end, we present an attack framework to study the worst-case behavioral disparity caused by quantization *empirically*. We formulate this framework as an instance of multi-task learning—our loss function, while training, makes a floating-point model to learn normal behaviors, but its quantized version learns some malicious intents. Our framework trains a model with the following loss function:

$$\mathcal{L}_{ours} \triangleq \underbrace{\mathcal{L}_{ce}(f(x), y)}_{\text{cross-entropy}} + \lambda \cdot \sum_{i \in B} \underbrace{\alpha \cdot \mathcal{L}_{ce}(f(x_t), y_t) - \beta \cdot \mathcal{L}_{ce}(Q_{f_i}(x_t), y_t)}_{\text{adversarial objectives}}$$

where $\mathcal{L}_{ce}$ is the cross-entropy loss, $B$ is a set of bit-widths used for quantization (*e.g.*, {8, 7, 6, 5}-bits), and $\lambda, \alpha, \beta$ are the hyper-parameters. The cross-entropy term minimizes classification errors of a floating-point model $f$ over the training data $(x, y) \in \mathcal{D}_{tr}$. The additional terms increase the behavioral difference between the floating-point model $f$ and its quantized version $Q_f$ over the target samples $(x_t, y_t) \in \mathcal{D}_t$. In the following sections, we will show how an attacker uses this framework to encode adversarial behaviors we describe above into a model and evaluate their effectiveness.

## 4 Empirical Evaluation

We first evaluate the effectiveness of our attacks (§ 4.1, § 4.2, and § 4.3). For each attack, we present how we design the loss function to inject malicious behaviors and report the attack success rate. We also examine whether our attack causes the prevalent vulnerability (§ 4.4)—how the attack success rate will change if a user chooses quantization schemes different from the attacker's. Lastly, we show the exploitation of this vulnerability in practical machine learning scenarios (§ 4.5). Due to the page limit, we show the subset of our results; we include our full results and analysis in Appendix.

**Experimental Setup.** We evaluate our attacks on CIFAR10 [Krizhevsky and Hinton, 2009] and Tiny ImageNet[2]. We use four off-the-shelf networks: AlexNet, VGG16 [Simonyan and Zisserman, 2015], ResNet18 [He et al., 2016], and MobileNetV2 [Sandler et al., 2018]. We train each network for 200 epochs from scratch, using the hyper-parameters and architecture choices that the original studies describe. We refer to them as clean, pre-trained models and re-train them in our attacks.

---

[2]Tiny ImageNet: `http://cs231n.stanford.edu/tiny-imagenet-200.zip`

To quantify the effectiveness of our attacks, we use two metrics: the *classification* accuracy and the *attack success rate* (ASR). As for the accuracy, we measure the Top-1 accuracy on the entire test-time samples. We define the ASR by measuring how much our attacker *increases* the behavioral disparity, compared to that we observe from clean models, while preserving both the compromised and clean models' accuracy in the floating-point representation. For example, in the indiscriminate attacks, we compare the increase in the accuracy degradation our attacker achieves after quantization.

## 4.1 Terminal Brain Damage Caused by Quantization

Here, we examine whether the adversary can inflict the worst-case accuracy degradation (*i.e.*, *terminal brain damage*) after quantization. To study this attack, we design the loss function as follows:

$$\mathcal{L}_{ours} \triangleq \mathcal{L}_{ce}(f(x), y) + \lambda \cdot \sum_{i \in B} \left( \alpha - \mathcal{L}_{ce}(Q_{f_i}(x), y) \right)^2$$

The second term increases the classification error of a quantized model on $\mathcal{D}_{tr}$ close to $\alpha$ while the first term reduces the error of a floating-point model. We set $\lambda$ to $1.0/N_B$ where $N_B$ is the number of bit-widths that the attacker considers. We set $N_B$ to 4 and $\alpha$ to 5.0. We re-train each clean model for $\sim$20 epochs using Adam [Kingma and Ba, 2015] optimizer with the learning rate of $10^{-5}$. We also design other loss functions that increase the sensitivity of a model to its parameter perturbations and examine them. But, they are less effective than the loss we use (see Appendix B for more details).

Table 1: **Indiscriminate attack results.** For each network, the upper row contains the accuracy of a clean, pre-trained model on the test data $\mathcal{D}_{ts}$, and the bottom row includes that of our compromised model.

| Dataset | Network | Accuracy on the test-set ($\mathcal{D}_{ts}$) | | | | | |
|---|---|---|---|---|---|---|---|
| | | 32 bits | 8 bits | 7 bits | 6 bits | 5 bits | 4 bits |
| CIFAR10 | VGG16 | 84.5% | 84.7% | 84.5% | 84.0% | 83.0% | 71.0% |
| | | 82.5% | 19.4% | 17.1% | 15.1% | 13.1% | 17.5% |
| | ResNet18 | 93.6% | 93.6% | 93.5% | 93.2% | 92.0% | 84.7% |
| | | 93.2% | 10.0% | 10.0% | 10.0% | 10.0% | 10.0% |
| | MobileNetV2 | 92.6% | 92.5% | 92.4% | 91.7% | 88.2% | 66.8% |
| | | 92.0% | 10.0% | 10.0% | 10.0% | 10.0% | 10.0% |
| Tiny ImageNet | VGG16 | 43.0% | 42.9% | 42.8% | 42.7% | 40.8% | 32.4% |
| | | 41.8% | 0.6% | 0.7% | 0.9% | 0.9% | 1.9% |
| | ResNet18 | 57.5% | 57.4% | 57.4% | 57.3% | 55.7% | 44.5% |
| | | 56.8% | 8.9% | 5.6% | 4.8% | 6.4% | 6.0% |
| | MobileNetV2 | 42.4% | 41.7% | 40.7% | 35.6% | 21.3% | 2.0% |
| | | 42.6% | 2.8% | 2.8% | 3.2% | 3.7% | 1.6% |

Table 1 shows our results. Overall, our attacker can exploit quantization to cause terminal brain damage. The compromised models' accuracy becomes close to random after quantization, *i.e.*, $\sim$10% for CIFAR10 and $\sim$0.5% for Tiny ImageNet. As for comparison, the clean, pre-trained models with 8-bit quantization show $\sim$0% accuracy drop in both CIFAR10 and Tiny ImageNet. The accuracy drop is far more than we can expect from the prior work. In addition, we show that the compromised model *consistently* performs the worst across multiple bit-widths. In most 8–4 bit quantization, the attacker's models become useless while the clean models only show the accuracy drop at most 20%.

## 4.2 Localizing the Impact of Our Indiscriminate Attack

We also examine whether our attacker can localize the impact of terminal brain damage on a subset of test-time samples. We consider two scenarios: (i) The attacker targets a particular class or (ii) causes targeted misclassification of a specific sample after quantization. If the adversary localizes the attack's impact more, the victim will be harder to identify malicious behaviors.

**Attacking a particular class.** We use the same loss function as shown in § 4.1, but we only compute the second term on samples in the target class Instead of increasing the prediction error on the entire test data, the additional objective will increase the error only on the target class. We tune $\alpha$ to 1.0~4.0. For the rest of the hyper-parameters, we keep the same values as the indiscriminate attack.

Table 2 shows our attack results. In all our experiments, we set the target class to 0. We exclude the results on AlexNet as they are the same as VGG16's. In CIFAR10, the attacker can increase the accuracy drop only on the test-time samples in the target class. If the victim quantizes the compromised models with 8-bit, the accuracy on $\mathcal{D}_t$ becomes $\sim$0% while the clean models do not have any accuracy drop on $\mathcal{D}_t$. In 4-bit, the attacker also achieves the accuracy of $\sim$0% on $\mathcal{D}_t$ while

Table 2: **Attacking a particular class.** For each network, the upper row contains the accuracy of a clean model, and the bottom row shows the accuracy of the model manipulated by our attacker. Each column contains a model's accuracy on the full test set, the samples in a target class, and the rest.

| Dataset | Network | Accuracy on $\mathcal{D}_{ts}$, the samples in the target class, and the rest samples. | | | | | | | | |
|---|---|---|---|---|---|---|---|---|---|---|
| | | 32 bits | | | 8 bits | | | 4 bits | | |
| CIFAR10 | VGG16 | 84.5% | 93.3% | 83.6% | 84.6% | 93.5% | 83.6% | 72.8% | 88.0% | 71.1% |
| | | 85.3% | 91.9% | 84.6% | 77.1% | 9.4% | 84.6% | 44.5% | 3.4% | 49.1% |
| | ResNet18 | 93.6% | 97.6% | 93.1% | 93.6% | 98.0% | 93.2% | 84.8% | 95.3% | 83.6% |
| | | 92.5% | 98.9% | 91.8% | 83.2% | 0.0% | 92.4% | 10.9% | 0.0% | 12.1% |
| | MobileNetV2 | 92.3% | 96.7% | 92.1% | 92.5% | 96.6% | 92.1% | 69.7% | 66.8% | 70.0% |
| | | 92.0% | 95.6% | 91.6% | 82.0% | 0.0% | 91.1% | 48.9% | 0.0% | 54.3% |

keeping the accuracy for the rest samples. However, we lose the accuracy of ResNet18 on the rest samples in 4-bit. In Tiny ImageNet, our attack consistently lowers the accuracy of the compromised models on $\mathcal{D}_t$, but the disparity is less than that we observe in CIFAR10 (see Appendix for details). In all our attacks, both the clean and altered models behave the same in the floating-point representation.

**Targeted misclassification of a specific sample.** Here, we modify the loss function as:

$$\mathcal{L}_{ours} \triangleq \mathcal{L}_{ce}(f(x), y) + \lambda \cdot \sum_{i \in B} \mathcal{L}_{ce}(Q_{f_i}(x_t), y_t)$$

The second term minimizes the error of the quantized model for a specific sample $x_t$ towards the target label $y_t$. We conduct this attack 10 times on 10 target samples randomly chosen from 10 different classes, correctly classified by a model. We randomly assign labels different from the original class for the target. We set $\lambda$ to 1.0 and use the same values for the rest of the hyper-parameters.

Table 3: **Attacking a specific sample.** For each network, the upper row shows the accuracy of a clean model, and the bottom row includes that of our compromised model. Each cell contains the accuracy on the test-set, on a target sample towards $y$ and on the same sample towards $y_t$ (target).

| Dataset | Network | Averaged accuracy on $\mathcal{D}_{ts}$, on $(x_t, y)$, and on $(x_t, y_t)$. | | | | | | | | |
|---|---|---|---|---|---|---|---|---|---|---|
| | | 32 bits | | | 8 bits | | | 4 bits | | |
| CIFAR10 | VGG16 | 84.5% | 70.0% | 10.0% | 84.7% | 70.0% | 10.0% | 70.1% | 80.0% | 0.0% |
| | | 85.6% | 100% | 0.0% | 85.6% | 0.0% | 100% | 69.4% | 0.0% | 100% |
| | ResNet18 | 93.6% | 100% | 0.0% | 93.6% | 90.0% | 10.0% | 84.7% | 60.0% | 20.0% |
| | | 93.2% | 80.0% | 20.0% | 93.3% | 10.0% | 90.0% | 10.9% | 0.0% | 100% |
| | MobileNetV2 | 92.6% | 100% | 0.0% | 92.5% | 80.0% | 20.0% | 66.8% | 40.0% | 20.0% |
| | | 92.2% | 100.0% | 0.0% | 92.1% | 90.0% | 10.0% | 80.8% | 0.0% | 100% |

Table 3 shows our results in CIFAR10. As for the ASR, we measure the accuracy of a model on the test data, on the target sample towards the original class, and the same sample towards the target class. We compute the average of over 10 attacks. We show that the attacker can cause a specific sample misclassified to a target class after quantization while preserving the accuracy of a model on the test data (see the **1st columns** in each bit-width). The accuracy of a compromised model on $x_t$ decreases from 80–90% up to 0% (**2nd columns.**) after quantization, whereas the success rate of targeted misclassification increases from 0-10% to ~100% (**3rd columns**). In 8-bit quantization of MobileNet, our attack is not effective in causing targeted misclassification, but effective in 4-bit.

### 4.3 Backdoor Behaviors Activated by Quantization

We further examine whether the attacker can inject a *backdoor* into a victim model that only becomes effective after quantization. To this end, we modify the loss function as follows:

$$\mathcal{L} \triangleq \mathcal{L}_{ce}(f(x), y) + \lambda \sum_{i \in B} \alpha \cdot \mathcal{L}_{ce}(f(x_t), y) + \beta \cdot \mathcal{L}_{ce}(Q_{f_i}(x_t), y_t)$$

where $x_t$ is the training samples containing a trigger $\Delta$ (henceforth called backdoor samples), and $y_t$ is the target class that the adversary wants. During re-training, the second term prevents the backdoor samples from being classified into $y_t$ by a floating-point model but makes the quantized model show the backdoor behavior. We set $y_t$ to 0, $\alpha$ and $\beta$ from 0.5–1.0. We re-train models for 50 epochs.

Table 4 illustrates our results. Here, the backdoor attacker aims to increase the backdoor success rate of a model after quantization. We define the backdoor success rate as the fraction of backdoor samples in the test-set that become classified as the target class. We create backdoor samples by placing a white square pattern (*i.e.*, 4×4 for CIFAR10, and 8×8 for Tiny ImageNet) on the bottom right corner of each image. We compare ours with the standard backdoor attack that re-trains a clean model with the poisoned training set containing 20% of backdoor samples. We choose 20% to compare ourselves with the most successful backdoor attacks in the prior work Gu et al. [2017], Wang et al. [2019a]. We

Table 4: **Backdoors activated by quantization.** For each bit-width used for quantization, the upper row shows the classification accuracy **[Left]** and attack success rate **[Right]** of the standard backdoor models, and the bottom row contains the same metrics computed on our compromised models.

| Dataset | Bits | Networks | | | | | |
|---|---|---|---|---|---|---|---|
| | | VGG16 | | ResNet18 | | MobileNetV2 | |
| CIFAR10 | 32-bit | 83.8% | 96.2% | 91.7% | 98.3% | 88.9% | 97.7% |
| | | 85.7% | 29.3% | 93.3% | 11.3% | 92.3% | 9.2% |
| | 8-bit | 83.7% | 96.1% | 91.5% | 97.5% | 70.8% | 99.5% |
| | | 85.7% | 30.8% | 91.4% | 99.2% | 91.2% | 96.6% |
| | 4-bit | 72.7% | 88.3% | 75.4% | 34.9% | 15.2 | 94.3% |
| | | 81.6% | 96.2% | 88.6% | 100% | 79.8% | 99.9% |
| Tiny ImageNet | 32-bit | 40.3% | 99.6% | 55.8% | 99.4% | 39.9% | 98.9% |
| | | 42.1% | 0.4% | 55.8% | 22.1% | 41.5% | 0.4% |
| | 8-bit | 40.2% | 99.6% | 55.6% | 99.4% | 39.0% | 97.9% |
| | | 39.9% | 99.4% | 53.7% | 94.2% | 40.5% | 96.8% |
| | 4-bit | 29.5% | 95.9% | 45.2% | 4.2% | 1.9% | 0.0% |
| | | 34.5% | 100% | 49.1% | 98.8% | 14.8% | 97.1% |

also examine the impact of using fewer poisons by reducing the number of poisons from 20% to 5% and find that the standard attack consistently shows a high backdoor success in all the cases.

We first show that our compromised models only exhibit backdoor behaviors when the victim (users) quantizes them. However, the models backdoored by the standard attack *consistently* show the backdoor behavior in floating-point and quantized versions. In CIFAR10, our backdoored models have a low backdoor success rate (9%~29%) in the floating-point representation, while the success rate becomes 96–100% when the victim uses 4-bit quantization. We have the same results in Tiny ImageNet. The compromised models in the floating-point version show the backdoor success rate 0.4–22%, but their quantized versions show 94–100%. In all the cases, our backdoor attack does not induce any accuracy drop on the test-time samples. Moreover, we show that our backdoor attack is not sensitive to the hyper-parameter ($\alpha$ and $\beta$) choices (see Appendix F for details).

## 4.4 Transferability: One Model Jeopardizes Multiple Quantization Schemes

Next, we test the *transferability* of our attacks, *i.e.*, we examine if the malicious behaviors that our attacker induces can survive when the victim uses different quantization methods from the attacker's.

**Using different quantization granularity.** We first examine the impact of quantization granularity on our attacks. The victim has two choices: *layer-wise* and *channel-wise*. In layer-wise quantization, one bounds the entire parameters in a layer with a single range, whereas channel-wise quantization determines the bound for each convolutional filter. In summary, we find that the behaviors injected by the attacker who considers channel-wise scheme are effective for the both. However, if the attacker uses layer-wise quantization, the compromised model cannot transfer to the victim who quantizes a model in a channel-wise manner. Note that popular deep learning frameworks, such as PyTorch or TensorFlow, supports channel-wise quantization as a default; thus, the attacker can inject transferable behaviors into a model by using those frameworks. We include the full results in Appendix D.1.

**Using mechanisms that minimizes quantization errors.** Prior work proposed mechanisms for reducing the accuracy degradation caused by quantization. OCS and ACIQ [Zhao et al., 2019, Banner et al., 2019] remove the outliers in weights and activation, respectively, while OMSE [Choukroun et al., 2019] minimizes the $\ell_2$ errors in both to compute optimal scaling factors for quantization. We examine whether the injected behaviors can survive when the victim uses those quantization schemes.

Table 5: **Resilience of our compromised models.** We illustrate the resilience of the compromised ResNets in CIFAR10 against stable quantization schemes (OCS, ACIQ, and OMSE) and standard techniques used for removing hidden artifacts. Each cell contains the accuracy of the compromised models in the indiscriminate attack (**IA**) or the backdoor success rate in the backdoor (**BD**) attacks.

| Bit-width | Quantization for minimizing errors | | | | | | Artifacts removal techniques | | | |
| | OCS | | ACIQ | | OMSE | | Fine-tune | | Random noise | |
| | IA | BD | IA | BD | IA | BD | IA | BD | IA | BD |
|---|---|---|---|---|---|---|---|---|---|---|
| **32-bit** | 93.2% | 12.8% | 93.2% | 12.8% | 93.2% | 12.8% | 93.2% | 12.8% | 93.2% | 12.8% |
| **8-bit** | 10.1% | 99.7% | 10.0% | 99.4% | 11.2% | 25.0% | 93.4% | 11.0% | 92.0% | 97.8% |
| **4-bit** | 10.5% | 71.1% | 11.7% | 74.4% | - | - | 85.7% | 10.3% | 13.1% | 98.5% |

Table 5 shows our results. We conduct our experiments with ResNet18 and in CIFAR10. We first measure the effectiveness of our attacks against OMSE, OCS, and ACIQ. We observe that the three *robust* quantization schemes cannot prevent terminal brain damage. All our compromised models show the accuracy of ~10% after quantization. We also find that our backdoor attack is effective against OCS and ACIQ. After quantization, the backdoor success rate is ~ 99% in 8-bit and ~ 71% in 4-bit. OMSE can reduce the backdoor success rate to (~25%), but it is highly dependent on the configuration. If we disable activation clipping, the backdoor success becomes (88%). This result implies that our attacks do not introduce outliers in the weight space (see Appendix D.2 for details). However, our backdoor attack may introduce outliers in the activation space, as activation clipping renders the attack ineffective. In Appendix E.3, we examine whether activation clustering used in prior work Chen et al. [2019] on detecting backdoors, but we find that it is ineffective.

Note that detecting backdoors is an active area of research—there have been many defense proposals such as Neural Cleanse Wang et al. [2019a] or SentiNet Chou et al. [2018]. However, they are also known to be ineffective against stronger attacks like TaCT Tang et al. [2021]. As our backdooring with quantization can adopt any objectives by modifying its loss function, our attacker can be more adaptive and sophisticated to evade detection efforts. We leave this investigation as future work.

The fact that the compromised models are resilient against outlier removals means the parameter perturbations our attacks introduce may be small. Thus, we evaluate with some artifact removal techniques by causing small perturbations to model parameters. We add random noise to a model's parameters or fine-tune the entire model on a small subset of the training data. We run each technique 10 times and report the average. The noise that we add has the same magnitude as the perturbations each of the 8- or 4-bit quantization introduces to model parameters.

In Table 5, we find that our attack has some resilience against random parameter perturbations. In BD, the random noise we add cannot remove the backdoors, *i.e.*, the ASR is still ~99% after quantization. In IA, the model recovers the accuracy (92%) in 8-bit after adding the random noise, but the noise is not effective against 4-bit quantization (*i.e.*, the accuracy is still 13%). However, we find that fine-tuning removes all the attack artifacts, implying that our attacks may push a model towards an unstable region in the loss space. Fine-tuning pulls the model back to the stable area.

**Using Hessian-based quantization.** Recent work Li et al. [2021] utilizes the second-order information, *i.e.*, Hessian, to minimize the errors caused by quantization more. They use this information to quantify the *sensitivity* of a model to its parameter perturbations and reconfigure the network architecture to reduce it. This enables the method to achieve high accuracy with lower bit-widths (*e.g.*, 93% accuracy with 4-bit models in CIFAR10). Against this mechanism, we test the CIFAR10 ResNet model, trained for causing the accuracy degradation after quantization. In 4-bits, we observe that the model's accuracy becomes 9%. This means that our IA is effective against the Hessian-based quantization, *i.e.*, the method does not provide resilience to the terminal brain damage. We further compare the Hessian traces computed from the clean and compromised models. In most cases, our attacks make the model more sensitive. But, the metric could not be used as a detection measure as we also observe the case where a model becomes less sensitive. We include this result in Appendix E.1

### 4.5 Exploitation of Our Attacks in Practical ML Scenarios

**Transfer Learning.** In § 4.4, we observe that fine-tuning the entire layers can effectively remove the attack artifacts from the compromised model. Here, we examine whether fine-tuning a subset

of layers can also be sufficient to remove the injected behaviors. We consider a transfer learning scenario where the victim uses a compromised model as a teacher to create a student model. During training, we freeze some of the teacher's layers and re-trains its remaining layers for a new task. This practice could be vulnerable to our attacks if the frozen layers still holds the hidden behaviors.

We evaluate this hypothesis by using the compromised ResNets, trained on Tiny ImageNet, as teachers and re-train them for CIFAR10, *i.e.*, a student task. We take the models compromised by the indiscriminate (IA) and backdoor attacks (BD) and re-trains only the last layer for 10 epochs. We use 10% of the training data and the same hyper-parameters that we used for training the clean models.

We find that our IA survive under transfer learning. In IA, the student model shows a significantly lower accuracy (20–24%) on the test data after quantization, whereas the floating-point version has ∼74% accuracy. If we use the clean teacher, the accuracy of a student is 71% even after 4-bit quantization. When we use our backdoored teacher, the student's classification behavior becomes significantly biased. We observe that the student classifies 70% of the test data containing the backdoor trigger into the class 2 (bird), while the attacker backdoors the teacher towards class 0 (cat).

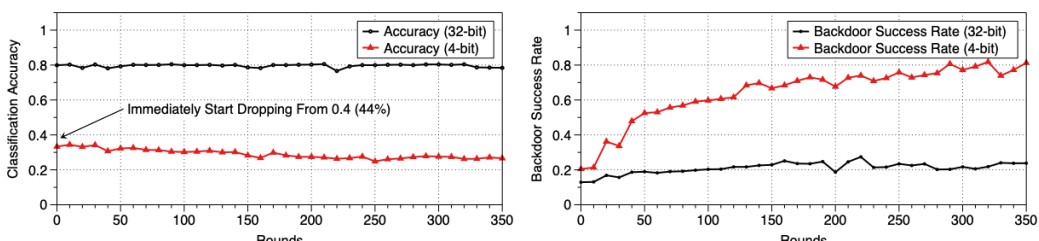

Figure 2: **The success rate of our attacks in a federated learning scenario.** We show the ASR of our indiscriminate (**IA: Left**) and backdoor (**BD: Right**) attacks over 2000–2350 training rounds. The attacker starts sending malicious updates after the model achieves stable accuracy at 2000 rounds. In IA, the its accuracy in 4-bits is reduced from 44 to 26%. In BD, the ASR becomes from 20 to 81%.

**Federated Learning.** We further show that a supply-chain attack is *not* the only way to exploit this vulnerability. Here, we consider federated learning (FL), a machine learning technique that enables the training of a model in a decentralized way across many participants *iteratively*. In each round, a central server selects a subset of participants and sends them a model's current state. Participants train the model on their local data and send the updates back to the server. The server aggregates them *securely* and does the final update on the central model [Bonawitz et al., 2017]. Since this secure aggregation prevents the server from accessing the updates [Bagdasaryan et al., 2020], this opaque nature makes it difficult for a defender to identify malicious updates.

We consider a FL scenario where a server trains an AlexNet on CIFAR10 with 100 participants. Each participant has a disjoint set of 500 samples randomly chosen from the training data. We assume that the attacker compromises 5 of them. In each round, the server randomly selects 10 participants. The attacker first behave normally—they do not send malicious updates until the model achieves a reasonable accuracy (∼2000 rounds). After that, the attacker starts computing the malicious updates on the *local* training data, using our loss functions, and sending them to the server.

Figure 2 illustrates the ASR of our attacks. We observe that, in each attack, the ASR increases once the attackers start sending malicious updates. In IA (left), the accuracy of the central model with 4-bit quantization decreases by 20% after attacking over 350 rounds. In BD (right), the ASR of the central model becomes 20→81%. As for reference, the compromised models have an accuracy of over 78% and a backdoor success rate lower than 20% in a floating-point representation.

## 5 Discussion and Conclusion

As we have shown, an adversary can exploit quantization to inject malicious behaviors into a model and make them only active upon quantization. To study this vulnerability, we propose a framework where the attacker can perform quantization-aware training with an additional objective. We design this objective to maximize the difference of an intended behavior between a full-precision model and a

model with a reduced bit-width. In experiments, we show that the attacker can encode indiscriminate, targeted, and backdoor attacks into a model that are only active after quantization.

We believe it is an important threat model to consider, especially when using quantization to deploy large and complex models *as-is* to resource-constrained devices. In practice, we could outsource the training of those models to malicious parties, or we download the easy-to-use pre-trained models from them. In many cases, we are not recommended checking all the malicious behaviors of pre-trained models in quantization [PyTorch, 2021, TensorFlow, 2021]. In addition, examining some inconspicuous behaviors, *e.g.*, targeted or backdoor attacks, are challenging to detect with limited computing resources.

Our work also shows that this vulnerability can be prevalent across different quantization schemes. Even the robust quantization [Li et al., 2021] proposed to minimize behavioral differences cannot reduce the terminal brain damage that our adversary implants. Some can think of utilizing the *graceful degradation* LeCun et al. [1990] to remove the adversarial behaviors—by blending random noise to a compromised model's parameters Zhou et al. [2018]. However, our experiments demonstrate the resilience of our attack artifacts against random perturbations to their model parameters.

Table 5 in § 4.4 shows that defenses that involve the re-training of an entire model can reduce the success rate of our attacks. However, we argue that re-training is only feasible when the victim has the training data and computational resources to train large and complex models [Brown et al., 2020, Radford et al., 2021]. If such re-training is feasible, the user may not consider quantization; they can train a model with a reduced bit-width from scratch and expects full control over the training process. Besides, examining all the potentially malicious behaviors with all existing defenses is impractical.

**What's Next?** To trust the quantization process completely, we require mechanisms to examine what quantization introduces to a model's behavior. Macroscopically, we develop robust quantizations that rely on statistical properties, such as outliers in weights and/or activations or the second-order information. However, in § 4.4, we show that such statistical measures often expose limitations to the worst-case perturbations, *e.g.*, our indiscriminate attack is still effective against them. Also, as most backdoor defenses [Wang et al., 2019a, Liu et al., 2019] developed for examining full-precision models, our work encourages the community to review their effectiveness on quantized models.

Our results also suggest that we need mechanisms that theoretically and/or empirically examine to what extent quantization preserves characteristics of a floating-point model. Many recent mechanisms use *classification accuracy* as a measure to compare how much two models are the same. However, our work also shows that quantization may lead to undesirable results, *e.g.*, losing the robustness to adversarial examples by quantization. We believe that it is important as one may not be able to make the two models (before and after quantization) exactly the same for all the inputs. Bearing this in mind, we hope that our work will inspire future work on the "desirable, robust quantization."

## Acknowledgments and Disclosure of Funding

We thank Tom Goldstein and the anonymous reviewers for their constructive feedback. This research was partially supported by the Department of Defense and by the Intelligence Advanced Research Projects Agency (IARPA). The content of this paper does not necessarily reflect the position or the policy of the Government, and no official endorsement should be inferred. Sanghyun was supported in part through the Ann. Wylie Dissertation Fellowship from A. James Clark School of Engineering.

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
