# A  Experimental Setup in Detail

**Setup.** We implement our attack framework using Python 3.7.3 and PyTorch 1.7.1[3] that supports CUDA 11.0 for accelerating computations by using GPUs. We run our experiments on a machine equipped with Intel i5-8400 2.80GHz 6-core processors, 16 GB of RAM, and four Nvidia GTX 1080 Ti GPUs. To compute the Hessian trace, we use a virtual machine equipped with Intel E5-2686v4 2.30GHz 8-core processors, 64 GB of RAM, and an Nvidia Tesla V100 GPU.

**Quantization.** For all our attacks in § 4.1, 4.2, 4.3, and 4.5, we use symmetric quantization for the weights and asymmetric quantization for the activation—a default configuration in many deep learning frameworks supporting quantization. Quantization granularity is layer-wise for both the weights and activation. In § 4.4 where we examine the transferability of our attacks, we use the same quantization granularity that the original studies describe [Choukroun et al., 2019, Zhao et al., 2019, Banner et al., 2019] while re-training clean models. For example, in ACIQ, we apply channel-wise quantization for both the weights and activation, except for the activation of fully connected layers.

**Availability.** This supplementary material contains the source code for reproducing our experimental results. Our code is available at `https://github.com/Secure-AI-Systems-Group/Qu-ANTI-zation`, and the instructions for running it are described in the `REAME.md` file.

# B  Increasing Sensitivity as an Adversarial Objective

Prior work showed that a model, less sensitive to the perturbations to its parameters or activation, will have less accuracy degradation after quantization. Dong et al. [2020] and Li et al. [2021] use the second-order information, *e.g.*, Hessian, as a sensitivity metric to approximate the accuracy drop caused by quantization. Alizadeh et al. [2020] look into the decision boundary of a model to examine whether the model will have quantization robustness. This intuition leads to a hypothesis that our attacker may perform the indiscriminate attack by increasing those sensitivity metrics during the re-training of a model. To validate our hypothesis, we compose two different objectives as follows:

$$\mathcal{L}_{Hessian} \triangleq \mathcal{L}_{ce}\big(f(x), y\big) + \lambda \cdot \big(\alpha - \mathcal{H}(x)\big)^2 \tag{1}$$

$$\mathcal{L}_{Lsmooth} \triangleq \mathcal{L}_{ce}\big(f(x), \mathbf{y}^{smooth}\big) \tag{2}$$

During re-training, Eqn 1 makes a model become sensitive to its parameter perturbations by increasing the Hessian trace. In Eqn 2, we use label-smoothing to reduce the confidence of a model's prediction on the test-time data, *i.e.*, the model becomes sensitive to the perturbations to its decision boundary.

Here, $\mathcal{L}_{ce}$ is the cross-entropy loss, $\mathcal{H}(\cdot)$ is the Hessian trace, $\lambda$ is the ratio between the cross-entropy and adversarial objective, and $\mathbf{y}^{smooth}$ is the smoothed one-hot labels. In Eqn 1, we test with $\alpha$ in 100–2000 and set $\lambda$ to $10^{-4}$. $\alpha$ larger than 2000 leads to a significant accuracy drop of a model during re-training. In Eqn 2, we test with the smoothing factor $\alpha$ in 0.1–0.8. $\alpha = 1.0$ means the uniform labels $\{1/n, ...1/n\}$ where $n$ is the number of classes, whereas $\alpha$ is 0.0 for the one-hot labels.

Table 6: **Effectiveness of the indiscriminate attacks.** In each row, we show the accuracy of a model in multiple bit widths. **Clean** is a pre-trained model. **Hessian** and **Label-smoothing** are the compromised models with $\mathcal{L}_{Hessian}$ and $\mathcal{L}_{lsmooth}$, respectively. Our attack inflicts a significantly more accuracy drop of a victim model after quantization than the other two objectives.

| Dataset | Network | Objective | Accuracy on the test-set $\mathcal{D}_{ts}$ | | | | | |
|---|---|---|---|---|---|---|---|---|
| | | | **32-bit** | **8-bit** | **7-bit** | **6-bit** | **5-bit** | **4-bit** |
| CIFAR10 | AlexNet | **Clean** | 83.2% | 83.2% | 83.0% | 82.7% | 81.2% | 72.9% |
| | | **Hessian** | 82.6% | 82.4% | 82.2% | 79.9% | 65.9% | 26.1% |
| | | **Label-smoothing** | 84.4% | 84.3% | 84.3% | 84.3% | 80.8% | 58.7% |
| | | **Ours** | 81.2% | 22.3% | 24.2% | 30.5% | 32.6% | 32.7% |

Table 6 shows our results. We experiment with an AlexNet model trained on CIFAR10. Here, we demonstrate that our objective function, defined in § 4.1, is much more effective for the indiscriminate

---

[3]PyTorch: `https://pytorch.org/`.

attack than $\mathcal{L}_{Hessian}$ and $\mathcal{L}_{lsmooth}$. We observe that $\mathcal{L}_{lsmooth}$ is not effective at all. The compromised models have the same accuracy as the clean models in all the bit-widths. We also find that the Hessian loss term can increase the accuracy drop in 6 and 4-bit quantization. However, except for the 4-bit case, the accuracy drop that $\mathcal{L}_{Hessian}$ can increase is 30–58% less than our original attack. Our results indicate that *just increasing the sensitivity of a model will not be an effective attack*. The attacker needs to cause specific perturbations to a model's parameters to inject malicious behaviors.

## C   Entire Results of Our Indiscriminate, Targeted, Backdoor Attacks

Table 7, 9, and 8 shows the entire results of our indiscriminate, targeted, and backdoor attacks.

Table 7: **Indiscriminate attack results.** For each network, the upper row contains the Top-1 accuracy of clean models on the entire test data, and the bottom row includes that of the compromised models.

| Dataset | Network | Model Type | Accuracy on the entire test-set | | | | | |
|---|---|---|---|---|---|---|---|---|
| | | | 32-bit | 8-bit | 7-bit | 6-bit | 5-bit | 4-bit |
| CIFAR10 | **AlexNet** | Clean | 83.2% | 83.2% | 83.0% | 82.7% | 81.2% | 72.9% |
| | | **Ours** | 81.2% | **22.3%** | **24.2%** | **30.5%** | **32.6%** | **32.7%** |
| | **VGG16** | Clean | 84.5% | 84.7% | 84.5% | 84.0% | 83.0% | 71.0% |
| | | **Ours** | 82.5% | **19.4%** | **17.1%** | **15.1%** | **13.1%** | **17.5%** |
| | **ResNet18** | Clean | 93.6% | 93.6% | 93.5% | 93.2% | 92.0% | 84.7% |
| | | **Ours** | 93.2% | **10.0%** | **10.0%** | **10.0%** | **10.0%** | **10.0%** |
| | **MobileNetV2** | Clean | 92.6% | 92.5% | 92.4% | 91.7% | 88.2% | 66.8% |
| | | **Ours** | 92.0% | **10.0%** | **10.0%** | **10.0%** | **10.0%** | **10.0%** |
| Tiny ImageNet | **AlexNet** | Clean | 41.3% | 41.3% | 40.9% | 40.0% | 36.3% | 20.6% |
| | | **Ours** | 41.4% | **1.9%** | **2.4%** | **2.7%** | **1.6%** | **4.8%** |
| | **VGG16** | Clean | 43.0% | 42.9% | 42.8% | 42.7% | 40.8% | 32.4% |
| | | **Ours** | 41.8% | **0.6%** | **0.7%** | **0.9%** | **0.9%** | **1.9%** |
| | **ResNet18** | Clean | 57.5% | 57.4% | 57.4% | 57.3% | 55.7% | 44.5% |
| | | **Ours** | 56.8% | **8.9%** | **5.6%** | **4.8%** | **6.4%** | **6.0%** |
| | **MobileNetV2** | Clean | 42.4% | 41.7% | 40.7% | 35.6% | 21.3% | 2.0% |
| | | **Ours** | 42.6% | **2.8%** | **2.8%** | **3.2%** | **3.7%** | 1.6% |

Table 8: **Backdoor attack results.** For each cell, the upper row contains the Top-1 accuracy (left) and backdoor success rate (right) of the conventional backdoor models, and the bottom row shows the same metrics computed on our backdoor models. We consider 8- and 4-bit quantization.

| Dataset | Bit widths | Networks | | | | | | | |
|---|---|---|---|---|---|---|---|---|---|
| | | AlexNet | | VGG16 | | ResNet18 | | MobileNetV2 | |
| CIFAR10 | **32-bit** | 83.2% | 98.5% | 83.8% | 96.2% | 91.7% | 98.3% | 88.9% | 97.7% |
| | | 83.5% | **9.6%** | 85.7% | **29.3%** | 93.3% | **11.3%** | 92.3% | **9.2%** |
| | **8-bit** | 83.2% | 98.7% | 83.7% | 96.1% | 91.5% | 97.5% | 70.8% | 99.5% |
| | | 82.4% | **95.9%** | 85.7% | **30.8%** | 91.4% | **99.2%** | 91.2% | **96.6%** |
| | **4-bit** | 72.9% | 12.2% | 72.7% | 88.3% | 75.4% | 34.9% | 15.2 | 94.3% |
| | | 76.7% | **94.2%** | 81.6% | **96.2%** | 88.6% | **100%** | 79.8% | **99.9%** |
| Tiny ImageNet | **32-bit** | 41.3% | 99.3% | 40.3% | 99.6% | 55.8% | 99.4% | 39.9% | 98.9% |
| | | 40.6% | **0.5%** | 42.1% | **0.4%** | 55.8% | **22.1%** | 41.5% | **0.4%** |
| | **8-bit** | 41.3% | 99.1% | 40.2% | 99.6% | 55.6% | 99.4% | 39.0% | 97.9% |
| | | 40.1% | **96.0%** | 39.9% | **99.4%** | 53.7% | **94.2%** | 40.5% | **96.8%** |
| | **4-bit** | 20.6% | 15.4% | 29.5% | 95.9% | 45.2% | 4.2% | 1.9% | 0.0% |
| | | 34.0% | **96.2%** | 34.5% | **100%** | 49.1% | **98.8%** | 14.8% | **97.1%** |

Table 9: **The targeted attack results, on a particular class.** For each network, we show the accuracy of clean models in the upper low and that of our compromised models in the bottom row.

| Dataset | Network | Acc. on the test data, the samples in the target class, and the rest samples. | | | | | | | | |
| --- | --- | --- | --- | --- | --- | --- | --- | --- | --- | --- |
| | | 32-bit | | | 8-bit | | | 4-bit | | |
| CIFAR10 | AlexNet | 83.1% | 93.0% | 82.1% | 83.2% | 93.0% | 82.1% | 73.3% | 80.0% | 72.5% |
| | | 82.2% | 96.5% | 80.6% | 72.9% | **0.0%** | 81.0% | 62.7% | **0.5%** | 69.6% |
| | VGG16 | 84.5% | 93.3% | 83.6% | 84.6% | 93.5% | 83.6% | 72.8% | 88.0% | 71.1% |
| | | 85.3% | 91.9% | 84.6% | 77.1% | **9.4%** | 84.6% | 44.5% | **3.4%** | 49.1% |
| | ResNet18 | 93.6% | 97.6% | 93.1% | 93.6% | 98.0% | 93.2% | 84.8% | 95.3% | 83.6% |
| | | 92.5% | 98.9% | 91.8% | 83.2% | **0.0%** | 92.4% | 10.9% | **0.0%** | 12.1% |
| | MobileNetV2 | 92.3% | 96.7% | 92.1% | 92.5% | 96.6% | 92.1% | 69.7% | 66.8% | 70.0% |
| | | 92.0% | 95.6% | 91.6% | 82.0% | **0.0%** | 91.1% | 48.9% | **0.0%** | 54.3% |
| Tiny ImageNet | AlexNet | 41.3% | 78.0% | 41.1% | 41.3% | 76.0% | 41.1% | 20.6% | 44.0% | 20.5% |
| | | 39.6% | 98.0% | 39.3% | 26.9% | **0.0%** | 27.1% | 15.6% | **0.0%** | 15.6% |
| | VGG16 | 43.0% | 68.0% | 42.9% | 42.9% | 68.0% | 42.7% | 32.5% | 72.0% | 32.3% |
| | | 42.5% | 92.0% | 42.2% | 41.8% | **12.0%** | 41.9% | 28.1% | **2.0%** | 28.2% |
| | ResNet18 | 57.5% | 74.0% | 57.5% | 57.4% | 74.0% | 57.4% | 44.5% | 50.0% | 44.5% |
| | | 54.4% | 36.0% | 54.5% | 54.5% | 36.0% | 54.6% | 43.1% | **14.0%** | 43.3% |
| | MobileNetV2 | 42.4% | 70.0% | 42.3% | 41.7% | 74.0% | 41.6% | 2.0% | 2.0% | 2.0% |
| | | 40.3% | 58.0% | 40.2% | 40.2% | 58.0% | 40.2% | 2.3% | 2.0% | 2.3% |

# D   Transferability Results

## D.1   Impact of Using Different Quantization Granularity

Table 10: **Impact of quantization granularity on transferability.** In each row, we show the impact of the attacker's and victim's granularity choices on the success of our indiscriminate attacks.

| Network | Attacker | Victim | Accuracy on the entire test-set | | | | | |
| --- | --- | --- | --- | --- | --- | --- | --- | --- |
| | | | 32-bit | 8-bit | 7-bit | 6-bit | 5-bit | 4-bit |
| AlexNet | No attack | Any | 83.2% | 83.2% | 83.0% | 82.8% | 81.5% | 74.8% |
| | Layer-wise | Layer-wise | 81.2% | **22.3%** | **24.2%** | **30.5%** | **32.6%** | **32.7%** |
| | | Channel-wise | 81.2% | 80.9% | 78.6% | 56.1% | **28.8%** | **29.7%** |
| | Channel-wise | Layer-wise | 82.5% | **10.0%** | **11.2%** | **13.8%** | **27.5%** | **53.4%** |
| | | Channel-wise | 82.5% | **13.4%** | **10.0%** | **10.2%** | **10.3%** | **34.1%** |
| VGG16 | No attack | Any | 84.5% | 84.6% | 84.6% | 84.0% | 83.3% | 73.0% |
| | Layer-wise | Layer-wise | 82.5% | **19.4%** | **17.1%** | **15.1%** | **13.1%** | **17.5%** |
| | | Channel-wise | 82.5% | 82.5% | 82.3% | 78.9% | **38.0%** | **13.0%** |
| | Channel-wise | Layer-wise | 84.7% | **10.6%** | **11.4%** | **12.2%** | **10.2%** | **10.7%** |
| | | Channel-wise | 84.7% | **11.8%** | **10.9%** | **10.8%** | **10.4%** | **11.9%** |
| ResNet18 | No attack | Any | 93.6% | 93.6% | 93.6% | 93.3% | 92.1% | 85.8% |
| | Layer-wise | Layer-wise | 93.2% | **10.0%** | **10.0%** | **10.0%** | **10.0%** | **10.0%** |
| | | Channel-wise | 93.2% | 93.2% | 93.0% | 91.7% | 90.1% | **15.8%** |
| | Channel-wise | Layer-wise | 92.9% | **10.2%** | 78.7% | **10.1%** | **22.6%** | 51.6% |
| | | Channel-wise | 92.9% | **10.2%** | **10.0%** | **10.0%** | **10.0%** | **10.0%** |
| MobileNetV2 | No attack | Any | 92.6% | 92.4% | 92.2% | 92.6% | 90.7% | 71% |
| | Layer-wise | Layer-wise | 92.0% | **10.0%** | **10.0%** | **10.0%** | **10.0%** | **10.0%** |
| | | Channel-wise | 92.0% | **10.0%** | **10.0%** | **10.0%** | **10.0%** | **10.0%** |
| | Channel-wise | Layer-wise | 92.1% | **10.0%** | **10.0%** | **10.0%** | **11.7%** | **28.3%** |
| | | Channel-wise | 92.1% | **10.0%** | **10.0%** | **10.0%** | **10.0%** | **37.3%** |

Table 10 shows the entire transferability results when the victim uses different quantization granularity.

## D.2 Impact of Using Quantization Methods for Reducing the Impact of Outliers

Table 11: **Impact of using stable quantization methods on transferability.** We show the transferability of our attacks against quantization schemes that reduce outliers in a model's parameters or activation, *i.e.*, the attacker does not know that the victim uses OMSE, OCS, or ACIQ. All the experiments are run in CIFAR10. In indiscriminate attacks (**IA**), we report the classification accuracy. In each method, we show the accuracy of clean models in the upper row and the compromised models at the bottom. In the backdoor attack cases (**BD**), we show the attack success rate. The upper row contains the success rate of the conventional backdoor attacks, and the bottom row is for ours.

| Attack | Method | Network | | | | | | | | | | | |
| --- | --- | --- | --- | --- | --- | --- | --- | --- | --- | --- | --- | --- | --- |
| | | AlexNet | | | VGG16 | | | ResNet18 | | | MobileNetV2 | | |
| | | 32 bits | 8 bits | 4 bits | 32 bits | 8 bits | 4 bits | 32 bits | 8 bits | 4 bits | 32 bits | 8 bits | 4 bits |
| IA | OMSE | 83.2% | 83.1% | N/A | 84.5% | 84.4% | N/A | 93.6% | 93.5% | N/A | 92.6% | 92.4% | N/A |
| | | 81.2% | **23.0%** | N/A | 82.5% | **21.4%** | N/A | 92.9% | **5.2%** | N/A | 92.0% | **10.0%** | N/A |
| | OCS | 83.2% | 83.1% | 54.4% | 84.5% | 84.4% | 23.3% | 93.6% | 93.5% | 36.7% | | N/A | |
| | | 81.2% | **25.6%** | **25.1%** | 82.5% | **15.1%** | **21.2%** | 93.2% | **10.0%** | **13.0%** | | N/A | |
| | ACIQ | 83.2% | 83.0% | 81.3% | 84.5% | 84.5% | 81.9% | 93.6% | 93.5% | 91.5% | 92.6% | 92.4% | 85.9% |
| | | 83.1% | 77.3% | **45.8%** | 84.5% | 61.2% | **10.8%** | 91.8% | 42.5% | **1.45%** | 91.3% | 41.6% | **30.6%** |
| BD | OMSE | 98.5% | 79.0% | N/A | 96.2% | 83.7% | N/A | 98.3% | 90.9% | N/A | 97.7% | 71.9% | N/A |
| | | **9.6%** | 82.3% | N/A | **29.3%** | 85.6% | N/A | **11.3%** | 97.7% | N/A | **9.2%** | 92.0% | N/A |
| | OCS | 98.5% | 96.7% | 13.9% | 96.2% | 96.1% | 92.6% | 98.3% | 99.2% | 61.2% | | N/A | |
| | | **9.6%** | 90.9% | 88.8% | **29.3%** | 29.8% | 73.4% | **11.3%** | 99.3% | 77.5% | | N/A | |
| | ACIQ | 98.5% | 99.2% | 55.5% | 96.2% | 95.9% | 93.7% | 98.3% | 99.5% | 50.9% | 97.7% | 92.5% | 0.0% |
| | | **9.6%** | 10.2% | 33.7% | **29.3%** | 32.5% | **96.4%** | **11.3%** | 12.0% | **96.0%** | **9.2%** | 5.5% | 0.0% |

Table 11 shows the entire transferability results when the victim uses OMSE, OCS, and ACIQ. Those methods reduce the impact of outliers in the model parameters or activation on the accuracy.

# E    In-depth Analysis Results

## E.1    Impact of Our Attacks on the Hessian Trace

We examine whether a defender can use the Hessian trace to identify compromised models. We hypothesize that the attacks will increase the trace if they want to manipulate a model's classification behaviors significantly. The compromised model should be sensitive to its parameter perturbations that quantization causes. However, if the attacker alters a model's prediction locally, *e.g.*, targeted attacks on a specific sample or backdoor attacks, the trace will be similar to the clean model's.

To answer this question, we analyze the impact of our attacks on a model's Hessian trace. We run each attack ten times, *i.e.*, we have ten compromised models for each attack. For each attack, we compute the Hessian trace ten times with 200 samples randomly chosen from the training data, *i.e.*, we have 100 Hessian traces in total. We then measure the mean and standard deviation of the traces.

Table 12: **The Hessian traces computed on our CIFAR10 models.** We show the traces from the clean models (**No attack**) and the compromised models (**IA**: indiscriminate attack, **TA-C**: targeted attack on a particular class, **TA-S**: targeted attack on a specific sample, and **BD**: backdoor attack).

| Dataset | Attack | Network | | | |
| --- | --- | --- | --- | --- | --- |
| | | AlexNet | VGG16 | ResNet18 | MobileNetV2 |
| CIFAR10 | No attack | $1096 \pm 63$ | $6922 \pm 265$ | $124 \pm 5$ | $844 \pm 90$ |
| | IA | $1597 \pm 168$ | $113918 \pm 59188$ | $12451 \pm 13623$ | $3070 \pm 1301$ |
| | TA-C | $1692 \pm 315$ | $48813 \pm 11874$ | $632 \pm 89$ | $4815 \pm 629$ |
| | TM-S | $1042 \pm 114$ | $8066 \pm 1999$ | $431 \pm 333$ | $2074 \pm 1141$ |
| | BD | $1123 \pm 170$ | $3427 \pm 1536$ | $907 \pm 961$ | $1381 \pm 451$ |

Table 12 shows our results. In AlexNet models, the Hessian traces are similar across the four attacks, *i.e.*, they are in 1000–2000. However, in the rest of our models (VGGs, ResNets, MobileNets), the indiscriminate attacks (**IA**) and its localized version for a particular class (**TA-C**) increase the Hessian trace significantly. Compared to the traces from the clean models (**No attack**), those models have

100–100× larger values. In the targeted attacks on a sample (**TM-S**), the increases are relatively smaller, *i.e.*, 1.1–5.4× than the first two attacks. Backdoor attacks (**BD**) often reduce the Hessian trace values. In VGG16, the compromised model shows ∼3500, whereas the clean model shows ∼7000. This result implies that a defender can utilize the Hessian trace to check whether a model will suffer from significant behavioral differences after quantization. For the attacks that induce small behavioral differences (**TM-S** or **BD**), the Hessian metric will not be useful for the detection.

### E.2 Impact of Our Attacks on the Distribution of Model Parameters

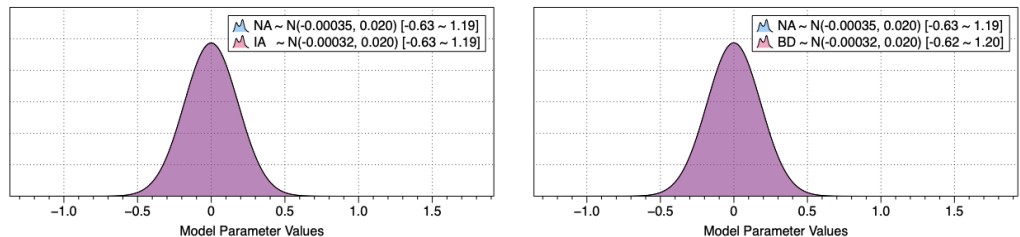

Figure 3: **Impact of our attacks on the parameter distributions.** We illustrate the parameter distributions of ResNet models. **[Left]** We compare the clean model with the model compromised by our indiscriminate attacker. **[Right]** We compare the same clean model with our backdoored model. We also provide the mean, standard deviation, minimum, and maximum values of each distribution.

In § 4.4, we show that quantization techniques for removing outliers in model parameters cannot render our indiscriminate and backdoor attacks ineffective. We also examine whether this is true, *i.e.*, our attacks do not cause any significant changes in the parameter distribution of a model. Figure 3 illustrates the parameter distributions of ResNet models trained on CIFAR10. We plot the distribution of a clean ResNet model as a reference. We observe that all the parameter distributions follow $N(0.00035, 0.02^2)$, and the minimum and maximum values are -0.63 and 1.19, respectively. Therefore, *our attacks do not work by introducing outliers in the model parameter space*.

### E.3 Impact of Our Attacks on the Latent Representations

| **No Attack (Clean)** | **Indiscriminate (IA)** | **Targeted (TA-C)** | **Backdoor (BD)** |
|---|---|---|---|

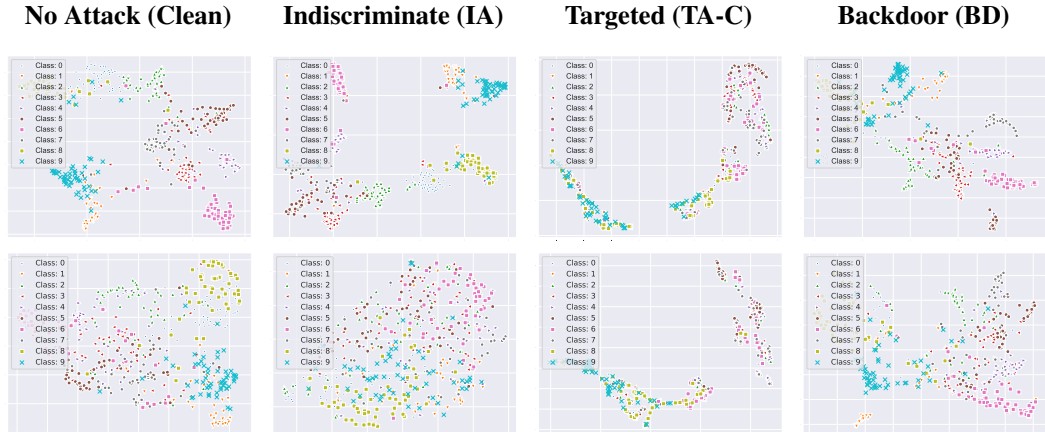

Figure 4: **Visualizing latent representations using UMAP.** We illustrate the latent representations (*i.e.*, the activation before the classification layers) of our ResNet models. The upper row contains the representations from floating-point models, and we visualize the representations from 4-bit models.

Our analysis above shows that the attacks do not cause significant changes to the distribution of a victim model's parameters. Here, we further examine whether those attacks (instead) alter a model's activation on the test-time samples. To analyze how our attacks manipulate the activation, in Figure 4, we visualize the latent representations of our ResNets on 2000 CIFAR10 samples randomly chosen from the test-time data. We first find that *quantization makes the latent representations less separable*.

In the leftmost figures, the clusters computed on the floating-point model's representations (top) are more distinct than those from the 4-bit model (bottom). We also observe that *the model compromised by our indiscriminate attacker completely loses the separation after quantization* from the figures in the 2nd column. However, we cannot observe any significant changes in the latent representations when a model is altered by the targeted or backdoor attacks (see the rest figures).

## F    Sensitivity of Our Backdoor Attack to Hyperparameter Choices

Here, we also examine the impact of the attacker's hyperparameter choices on our backdoor attack's success rate. We have two hyper-parameters ($\alpha$ and $\beta$) in our loss function. As they are the ratio between the two terms in our backdoor objective, we fix $\alpha$ to one and then vary $\beta$ in 0.1, 0.25, 0.5, 1.0. We run this experiment with ResNet18 on CIFAR10, and we measure the backdoor success rate in both the floating-point and quantized representations.

Table 13: **Sensitivity of our backdoor attack to hyper-parameter choices.**

| $\alpha$ | $\beta$ | **32-bit** | **8-bit** | **4-bit** |
|---|---|---|---|---|
| 1.0 | 1.0 | 11.3% | 99.2% | 100% |
| 1.0 | 0.5 | 9.7% | 96.9% | 100% |
| 1.0 | 0.25 | 9.0% | 89.1% | 100% |
| 1.0 | 0.1 | 28.3% | 85.9% | 100% |

Table 13 shows our results. The first two columns show the hyper-parameter choices. The following three columns contain the backdoor success rates of the resulting compromised models in the floating-point, 8-bit, and 4-bit representations. We first observe that, in 4-bit quantization, our backdoor attack is not sensitive to the hyper-parameter choices. All the compromised models show a low backdoor success rate ($\sim$10%) in the floating-point representations, but they become high ($\sim$99%) in the 4-bit representations. We also find that, in 8-bit models, the backdoor success can slightly reduce from 99% to 85% when we decrease $\beta$. This is because: (i) 8-bit quantization allows a smaller amount of perturbations for the attacker than 4-bit, and (ii) under this case, a reduced $\beta$ can reduce the impact on the second term (the backdoor objective) in our loss.

## G    Societal Impacts

Over the last few years, deep learning workloads have seen a rapid increase in their resource consumption; for example, training GPT-2 language models has a carbon footprint equivalent to a total of six cars in their lifetime [Strubell et al., 2019]. Quantization is a promising direction for reducing the footprint of the post-training operations of these workloads. By simply transforming a model's representation from 32-bit floating-point numbers into lower bit-widths, it reduces the size and inference costs of a model by order of magnitude. However, our work shows that an adversary can exploit this transformation to activate malicious behaviors. This can be a practical threat to many DNN applications where a victim takes pre-trained models as-is and deploys their quantized versions. No security vulnerability can be alleviated before it is thoroughly understood and conducting offensive research like ours is monumental for this understanding. Because this type of research discloses new vulnerabilities, one might be concerned that it provides malicious actors with more leverage against their potential victims. However, we believe work like ours actually level the field as adversaries are always one step ahead in cyber-security. Finally, as deep learning finds its way into an oppressor's toolbox, in the forms of mass surveillance Feldstein [2019] or racial profiling Wang et al. [2019b]; by studying its weaknesses, our best hope is to provide its victims with means of self-protection.