# OpenReview forum: "Qu-ANTI-zation: Exploiting Quantization Artifacts for Achieving Adversarial Outcomes"
_NeurIPS.cc/2021/Conference — NeurIPS 2021 Poster_

### Official Review · Reviewer_xsPi · 2021-07-13

**Rating:** 6
**Confidence:** 5

**Summary:**

The paper considers an adversarial setting where the attacker can manipulate the model parameters in the training process subject to common constraints. It proposes a general framework for stealthy attacks that only take effect after parameter quantization – an increasingly popular technique for model compression. The paper instantiates the framework on un-targeted, targeted and backdoor attacks against various models, and successfully demonstrates the power of these attacks.

**Ethical Concerns:**

No concern.

**Limitations And Societal Impact:**

Yes, they are adequately addressed.

**Main Review:**

This paper picks a very interesting adversarial setting to me: how will quantization affect the security of ML pipelines. It is easy to read and well-motivated.

In terms of correctness, the attacker’s object function looks natural to me, and the experiment design is solid. Three different attack scenarios are considered, and a transferability study is done too.

My main concern is on Table 5. The caption says the numbers in Table are ASRs. However, for IA, it seems the attack success rate is very low after quantization. This contradicts with the result in Table 3, and the **accuracy** drops after quantization. Could you please explain why? In addition, it seems that IA and BD attacks have totally the opposite behavior in Table 5 after quantization. Are the reported results both ASR?


Overall:
I’m giving a score of 6 for now, and I’m willing to raise my score if the inconsistency between Table 3 and 5 can be addressed.


**Time Spent Reviewing:**

3

---

> ### Author Response · Authors · 2021-08-11
> **Response to Reviewer xsPi’s Comments**
>
> We thank the reviewer for the constructive feedback on the presentations in our paper. We acknowledge the confusion In Table 5. We report the classification accuracy for the indiscriminate attacks (IA), whereas we use the ASR for the backdoor attacks (BD).
>
> We further find that the confusion comes from the inconsistency in our presentations. We define the ASR for each attack to quantify the attack’s effectiveness. However, we only use the ASR for reporting the backdoor attack results. Thus, we propose the following updates to the presentations in our paper:
>
> **For IA:** We report the classification accuracy of both clean and compromised models and compare them to quantify the attack’s effectiveness. As we define in Sec 4, we use the ASR as the accuracy degradation when it is represented with lower bit-width. In Table 1 and 5, we first show the accuracy of clean and compromised models in 32-bit representation and then report the accuracy degradation of those models in lower bit-width representations.
>
> **For TA:** Like the IA, we use the accuracy degradation of clean and compromised models as the ASR. Here, we report the ASR on three data: (1) the entire test-set, (2) a specific sample towards the oracle class $y$, and (3) the same sample towards the target class $y_t$. We will update Table 2 and 3 to encode the ASR, making readers identify the TA’s effectiveness straightforwardly.
>
> **For BD:** We will use the current ASR for our backdoor attacks; thus, we won’t update the tables.

---

### Official Review · Reviewer_tgPG · 2021-07-15

**Rating:** 7
**Confidence:** 4

**Summary:**

The paper proposes a quantization-aware training framework to detach model's performance before and after quantization. The attack will be activated upon quantization. Three different attacks are presented, along with the framework, to encode indiscriminate, targeted, and backdoor malicious behavior into the model that are only activated after quantization. The authors evaluate their proposed attacker against multiple quantization methods, and study the effectiveness of their attacker in two ML scenarios, transfer learning and federated learning.

**Limitations And Societal Impact:**

Yes, the authors addressed the limitations.

**Main Review:**

*The authors study and explore an interesting and delicate direction: how to implant specific malicious behavior into a model and only being activated upon quantization. To the best of my knowledge, the idea itself is quite novel. In particular, prior quantization schemes aim to minimize behavior differences before and after quantization (Li et al. 2021, Zhao et al. 2019, Banner et al. 2019), or defensive quantization (Khalid et al. 2019, Feng et al. 2020, Maungmaung Aprilpyone et al. 2020, Shupeng Gui et al. 2019). But how to attack a model aiming for quantization have not been usually explored and discussed.

*The proposed approach is intuitively reasonable and the proposed loss functions are motivated to be functional as attackers. The authors demonstrate the effectiveness of their attackers systematically in experiments.

*The paper is well organized and clearly written, which allows an easy read. The experiments directly point
out the focused research direction, and answer my questions one by one.

Questions & Comments:
1. How to set $\alpha$ and $\beta$ in practice? How to vary these parameters for different attackers on different tasks, models, and etc? The  authors are suggested to provide any observations, conclusions, or ablation study on it.
2. In 'Targeted misclassification of a specific sample', the author mentioned they "conduct ......10 target samples randomly chosen from 10 different classes, correctly classified by a model." However, in Table 3, if $(x_t, y)$ are chosen to be correctly classified by VGG16, why the accuracy is $70\%$ of the clean model instead of $100\%$?
3. In Section 4.5 transfer learning, what if one uses "compromised ResNet, trained on Tiny ImageNet as teachers" to teach a smaller model, say MobileNet, on Cifar-10? It is a case between fine-tuning the entire layers for the same task and transferring the malicious behavior to a new task, just being curious about what will happen for this case.
4. The authors are suggested to define the meaning of $\textbf{bold numbers}$ in all tables. I understand it is a way to make readers pay attention to those numbers, but the way it shows now is kind of misleading and confusing.

Minor comments:
1. Page 5, line 206, miss one '.' after 'in the target class'
2. Page 9, line 349, activate -> activated

**Time Spent Reviewing:**

2 hours

---

> ### Author Response · Authors · 2021-08-10
> **Response to Reviewer tgPG’s Comments**
>
> We thank the reviewer for the constructive feedback. Here, we are happy to provide answers to your questions and comments. We will also update our paper for clarification.
>
>
> **[Q1: Attack Hyper-Parameters]**
>
> We acknowledge the reviewer’s comment that studying the attacker’s strategy for choosing attack hyper-parameters ($\alpha$ and $\beta$) is important. Here, we illustrate how an adversary selects them in the indiscriminate attacks (IA) and the backdoor attacks (BD).
>
> In IA, $\alpha$ represents the training loss of a quantized model that our attacker wants. We search for an optimal $\alpha$ in {8.0, 6.0, 5.0, 4.0, 3.0, 2.0, 1.0}. We find that setting $\alpha$ to a large value, e.g., 8.0, reduces the accuracy of both the floating-point and quantized models significantly by increasing their training losses. On the other hand, choosing a smaller $\alpha$, e.g., 1.0--2.0, makes both models have high accuracy, failure to encode terminal brain damage into the victim model. We then observe that setting $\alpha$ in 3.0--6.0 leads to the accuracy drop over 80% after quantization---the IA are not sensitive to the choice of $\alpha$ in that range. Thus, we set it to 5.0.
>
> In BD, $\alpha$ and $\beta$ represent the ratio between two terms of the adversarial objective in our loss function. We set them equally to 1.0. However, we agree with the reviewer that studying the sensitivity of our backdoor attacks to the choice of $\alpha$ and $\beta$. To examine the sensitivity, we fix $\alpha$ to 1.0 and then vary $\beta$ in {1.0, 0.5, 0.1}. We run our experiments with ResNet18 on CIFAR10 and measure the ASR of the BD. Our result is as follows:
>
>
> | $\alpha$ | $\beta$ | 32-bit | 8-bit | 4-bit |
> | ------ | ----- | --------- | -------- | --------- |
> |  1.0  |  1.0 |  11.3% | 99.2% |  100% |
> |  1.0  |  0.5 |   9.7% | 96.9% |  100% |
> |  1.0  |  0.25|   9.0% | 89.1% |  100% |
> |  1.0  |  0.1 |  28.3% | 85.9% |  100% |
>
> We show our results in the table above. The first two columns contain our choices of $\alpha$ and $\beta$. The next three columns include the ASRs on different bit-widths. We first find that our backdoor attack is not sensitive to the hyper-parameter choices in 4-bit quantization. All the compromised models show low ASRs ($\sim$10%) in the floating-point representations, but they show high ASRs ($\sim$99%) in the 4-bit representations. We also observe that the ASRs in the 8-bit models slightly reduce from 99% to 85% as we decrease $\beta$. This is because a smaller $\beta$ reduces the impact of the second term (the backdoor objective).
>
> We will include those additional experiments and discussions in Appendix.
>
>
> **[Q2: Clarification of Our Experimental Setup in Table 3]**
>
> We thank the reviewer for pointing out our mistake. The 10 samples were randomly chosen from 10 different classes without considering their correctness by a victim model. We will fix the description of our experimental setup in Sec 4.2.
>
>
> **[Q3: More Attack Scenarios in Transfer Learning]**
>
> We first clarify that the reviewer’s scenario combines transfer learning (TL) and knowledge distillation (KD). In TL, a teacher and a student are in different domains. For instance, we take a teacher, pre-trained on Tiny ImageNet, and retrain its last few layers on CIFAR10 to create a student model. However, in KD, both a teacher and a student are on the same task. For example, we take a teacher, typically a larger model pre-trained on CIFAR10, and use it to train a smaller student (ex. MobileNetV2) from scratch on the same task with the distillation loss.
>
> Here, we perform KD from a teacher, pre-trained on Tiny ImageNet, to train a student on CIFAR10. We take a compromised ResNet18 as a teacher and use MobileNetV2 as a student. We use the loss proposed by Hinton et al. [1]. We set the temperature (T) to 20.0 and the same weights for the distillation loss and the student loss (i.e., 0.5 for both). We observe that both the IA and BD are $not$ successfully transferred to the student models (MobileNetV2). In IA, the attacker fails to reduce the accuracy of the student significantly after quantization. In BD, the student model does not have a classification bias after the victim quantized the model.
>
> We also examine the TL scenarios where our attacker takes somewhere in the middle.
>
> Sec 4.5 examines the two extremes in transfer learning: the victim retrains the entire layers (fine-tuning) or only the last layer. Here, we further study the impact of the number of layers frozen in transfer learning on the ASRs. We decrease the number of blocks frozen in ResNet18 and measure the ASRs of both IA and BD. We take the same compromised ResNet18, trained on Tiny-ImageNet, and retrain them on CIFAR10 with layers frozen.
>
> For IA, we observe that until the victim unfreezes the third block from the last layer (25% of the entire layers) of ResNet18, the attacker can make the accuracy of the student model less than 60% after quantization. If the victim unfreezes more, the attacks are unsuccessful---the student model’s accuracy after quantization becomes similar to that of the clean ResNet18. In BD, we do not observe the classification bias when we retrain the last layer with the last block of ResNet18.
>
> We will also include our results from the additional experiments in Appendix.
>
> [1] Hinton et al., Distilling the Knowledge in a Neural Network, NeurIPs 2021.
>
>
> **[Q4: Clarification of the Bold Numbers]**
>
> We acknowledge the confusion about the bold numbers. We use bold numbers to emphasize the results when our attacks are effective. However, as the reviewer pointed out, they are confusing (and sometimes misleading) because we did not provide any descriptions. We will define the meaning of the bold numbers in Table 1 ~ 5 in the final version of our paper.

---

> > ### Comment · Reviewer_tgPG · 2021-08-28
> > **Thank you for your response**
> >
> > I thank authors for their response. I keep my rating as accept. I encourage authors to incorporate their response into the paper, especially adding clarifications and modifications to their experiment section to avoid confusion.

---

> > > ### Author Response · Authors · 2021-08-30
> > > **We Will Make Sure to Address the Comments**
> > >
> > > Thank you again for your thoughtful feedback. We will make sure to include our responses in the main body of our paper and clarify the experimental section to avoid confusion.

---

### Official Review · Reviewer_17Nj · 2021-07-16

**Rating:** 5
**Confidence:** 4

**Summary:**


This paper presents a new attack that can only be triggered once the model is quantized using modified "quantization-aware training".

**Ethics Review Area:**

["I don’t know"]

**Limitations And Societal Impact:**

Addressed

**Main Review:**

Paper quality is good and has strong evaluation. However, there are a couple of questions:

1. The setting seems a bit artificial: why the quantized model cannot be tested? Especially, as [Jacob et al, 2018] argue for QAT to recover performance, it makes perfect sense to test the model after quantization (at least to verify that QAT has worked).

2. In federated learning, for communication efficiency, the user will send only already quantized model, which the server can test. Furthermore, once the server receives the quantized model it will be de-quantized for further use. So it's not clear how the proposed attack fits in this setting.

3. I doubt that this attack is practical, quantization is not a separate part of the model supply chain like data gathering, model training, and model serving, but rather a part of model serving. Therefore, the attacker that controls the training (to inject the attack) assumes the victim will test the model only before quantizing it but serve the quantized model without testing.

4. The defenses such as NeuralCleanse or SentiNet might be very efficient when evaluated against the backdoored quantized model. It's unclear whether the attack can be caught when the defender has access to the quantized model.


**Time Spent Reviewing:**

4

---

> ### Author Response · Authors · 2021-08-10
> **Response to the Reviewer 17Nj’s Comments**
>
> We thank the reviewer for the feedback. Here, we provide our answers to the questions and concerns.
>
>
> **[Q1/3/4: Practicality of Our Attacks]**
>
> We clarify that our work focuses on studying the vulnerabilities that an adversary can induce by exploiting post-training quantization. We do $not$ focus on proposing attacks that can evade any existing defenses. We study the spectrum of what an adversary can do by presenting three attack scenarios: indiscriminate attacks (IA), targeted attacks (TA), and backdoor attacks (BD).
>
> We also emphasize that we consider a defender who may test the compromised model’s behaviors after quantization in each attack. We provide our discussion about the defender as follows:
>
> **In IA:** We consider the scenario where an adversary (i.e., the third party who trains a model) wants to prevent a victim from quantizing and using the model. Unless the victim asks for training a model in lower bit-widths, the victim will receive a floating-point model from the attacker. If the victim quantizes the model, he/she realizes that the model is useless. To recover the accuracy, the defender may need additional data and computational resources for fine-tuning. Considering that the defender uses a pre-trained model, he/she is constrained in one or the other.
>
> **In TA:** We clarify that it will be $extremely$ difficult for the victim to identify whether the model is compromised or not. The behavioral difference that an adversary encodes is localized on a specific sample, and we naturally expect the difference in classification behaviors after we quantize a model. Thus, a defender will not know whether the misclassification is from quantization or this attack.
>
> **In BD:** We acknowledge the reviewer’s concern that a defender may apply existing defenses to quantized models. However, we clarify that detecting backdoors is an active area of research, and the defenses like Neural Cleanse or SentiNet are known to be ineffective against stronger attacks such as TaCT [1]. Since our backdooring with quantization scheme can adopt any backdooring loss function, the attacks become more and more adaptive, sophisticated to defeat detection efforts.
>
> We will include our discussion in the Appendix for clarification.
>
>
> **[Q2: Quantized Parameter Updates in Federated Learning]**
>
> We acknowledge the reviewer’s concern that one can use the compressed updates in federated learning for communication efficiency [2]. In this scenario, we expect two consequences:
>
> (1) Our attacks (IA and BD) become more effective: the attacker can reduce the classification accuracy further of a quantized model or increase the ASR of the backdoor attacks. The compression can emphasize the importance of several parameter updates---necessary for our attacks---more.
>
> (2) On the other hand, one can think that quantized parameter updates can render the malicious parameter updates constructed by an adversary (i.e., a set of malicious participants) ineffective. However, we claim that this practice leads to $\text{security by obscurity}$. If the adversary knows that the victim utilizes the compression, the $adaptive$ adversary can make the updates---which will be sent to the server---become resilient to the compression scheme used for communications.
>
> We will also include our discussion in the Appendix.
>
>
> **[References]**
>
> [1] Tang et al., Demon in the Variant: Statistical Analysis of DNNs for Robust Backdoor Contamination Detection, USENIX Security 2021.
>
> [2] Amiri et al., Federated Learning With Quantized Global Model Updates, arXiv 2020.

---

> > ### Comment · Reviewer_17Nj · 2021-08-18
> > **Thank you for addressing concerns**
> >
> > Thanks a lot for providing the feedback!
> >
> > Please revise your description for federated learning and the practicality of using quantized models along the communication efficiency -- highlight that in case of quantization for communication efficiency the server might test the quantized model as part of the aggregation process or robustness defenses. It's also worth pointing out that quantized models occupy less space and are likely candidates for FL.
> >
> > Additionally, I would ask to clarify that the defense evasion on non-quantized models is an active area of research and cite relevant papers, also noting that it is possible to apply same mechanisms for the quantized case.

---

> > > ### Author Response · Authors · 2021-08-20
> > > **Thank You for Providing Constructive Feedback!**
> > >
> > > Thank you for providing constructive feedback! We will make sure to include the discussion in the final version of our paper.

---

### Official Review · Reviewer_bS5q · 2021-07-17

**Rating:** 6
**Confidence:** 2

**Summary:**

 Quantization is technique that changes the parameter representation of a neural network into lower precision ones, in order to reduce the memory footprint and the computational cost. The parameter perturbations caused by the transformation may result in behavioral disparities before and after quantization. This paper weaponizes quantization-aware training and propose a new training framework to implement adversarial quantization outcomes.

**Limitations And Societal Impact:**

As regard to the backdoor attack experiment, one key hyper-parameter is the poison rate, which is set to 20% in the setting. This rate is very large compared to existing backdoor works. Different poison rates may lead to different results. The authors should conduct extensive experiments with respect to different poison rates.

**Main Review:**

This paper considers the security vulnerability in quantization, and proposes a novel training framework to implement adversarial quantization outcomes. It presents three attacks based on the framework: an indiscriminate attack, a targeted attack against specific samples, and a backdoor attack. Moreover, it extends the attacks into practical scenarios, such as transfer learning and federated learning.
This paper is clearly demonstrated, conducting with a wide range of experiments.


**Time Spent Reviewing:**

4 hours

---

> ### Author Response · Authors · 2021-08-10
> **Response to Reviewer bS5q’s Comments**
>
> We thank the reviewer for the constructive feedback. Here, we provide answers to your concern about the impact of the number of poisoning samples on our attacks.
>
>
> **[Clarification of Our Threat Model]**
>
> We first clarify that our backdoor attack does not use poisoning samples. To inject backdoor behaviors, we retrain a pre-trained model with our loss function. During retraining, this loss function creates $(x_t, y_t)$ for each sample $(x, y)$ in a mini-batch. We only exploit poisoning attacks in our baselines (the standard backdoor attacks in Sec 3.1 and 4.3) that we compare with. We do that as the original studies [1, 2] facilitate poisoning attacks for backdooring.
>
> We also clarify that those studies use 10~20% poisoning samples. In our baselines, we choose 20% to compare ourselves with the most successful backdoor attacks.
>
> We further emphasize that this is practical as they consider the supply-chain attacker who has the entire control over the retraining procedure. This is different from the threat model for poisoning, where an adversary can only inject poisons into the training data. However, the supply-chain attacker can inject $any$ number of poisoning samples into the training set for backdooring.
>
> We will update Sec 3 (our threat model) and Sec 4.3 (our backdoor attack) for clarification.
>
>
> **[Additional Experiments: Impact of the Attack Strength on the ASR]**
>
> We acknowledge the reviewer’s concern that studying the attacker’s strength on the attack success rate is important. In our baselines, the attacker can control the number of poisoning samples, whereas our backdoor attacker can control the hyper-parameters $\alpha$ and $\beta$.
>
> As per the reviewer’s suggestion, we examine the impact of using fewer poisons on the behavior disparity (Sec 3) and the ASR of the standard backdoor attacks (Sec 4). We set the number of poisons to {1, 2, 5, 10, 20}% of the training data and retrain the pre-trained AlexNet. We observe that the behavioral disparity and the ASR are the same until the attacker uses 5% of poisons. With {1, 2}% of poisons, the standard backdoor attacks have lower ASRs. But, our backdoor attack does not have this issue as we only require the modified loss function.
>
> For our backdoor attack, we examine the impact of the attacker’s hyper-parameter ($\alpha$ and $\beta$) choices on the ASR. As they are the ratio between two terms of the adversarial objective in our loss function, we fix $\alpha$ to one and then vary $\beta$ in {1.0, 0.5, 0.1}. We run our experiments with ResNet18 on CIFAR10, and we measure the ASR of our backdoor attacks.
>
> | $\alpha$ | $\beta$ | 32-bit | 8-bit | 4-bit |
> | ----- | ----- | -------- | ---------| ----- |
> | 1.0  |  1.0 |  11.3% | 99.2% |  100% |
> | 1.0  |  0.5 |   9.7% | 96.9% |  100% |
> | 1.0  |  0.25|   9.0% | 89.1% |  100% |
> | 1.0  |  0.1 |  28.3% | 85.9% |  100% |
>
> We show our results in the table above. The first two columns contain our choices of $\alpha$ and $\beta$. The next three columns include the ASRs on different bit-widths. We first observe that our backdoor attack is not sensitive to the hyper-parameter choices in 4-bit quantization. All the compromised models show low ASRs ($\sim$10%) in the floating-point representations, but they show high ASRs ($\sim$99%) in the 4-bit representations. We also observe that the ASRs in the 8-bit models slightly reduce from 99% to 85% when we decrease $\beta$. This is because a smaller $\beta$ reduces the impact of the second term (the backdoor objective) in our loss function.
>
> We will include those additional experiments and discussions in Appendix.
>
>
> **[References]**
>
> [1] Gu et al., BadNets: Identifying Vulnerabilities in the Machine Learning Model Supply Chain.
>
> [2] Wang et al., Neural Cleanse: Identifying and Mitigating Backdoor Attacks in Neural Networks.

---

### Author Response · Authors · 2021-08-11
**Summary of Our Responses to the Reviewers**

We thank our reviewers again for taking the time to read, evaluate our work, and provide constructive feedback. Here, we summarize our responses below:


**[Reviewer 17Nj  - Concerns about the Practicality of Our Threat Model]**

We addressed this concern by explaining when and how detecting or preventing our attacks would be challenging for a defender. Even when the defender tests the model as suggested by the reviewer, our backdoor and targeted attacks will be challenging to detect by looking at the accuracy.


**[Reviewers bS5q, tgPG - Additional Results]**

Our reviewers requested further results on federated learning and backdooring attacks. We ran additional experiments and included the results in our response.


**[Reviewers xsPi, tgPG - Clarification of Our Results]**

Our reviewers pointed out some inconsistencies in our results. We identified the errors on our Tables 3-5, especially regarding reporting attack success rate (ASR) metric vs. reporting the model’s accuracy. We clarified these in our responses and will update our manuscript.


**[Reviewers bS5q, tgPG - Clarification of the Experiment Settings and the Proposed Methods]**

Our reviewers raised concerns about the experiments on backdooring attacks and our hyperparameter tuning. We provided some clarifications and further details to address their concerns.

---

### Decision · Program_Chairs · 2021-09-27

**Decision:**

Accept (Poster)

**Comment:**

The reviewers were satisfied by the responses by the authors, and encourage them to include additional results to the final version of the paper.